# Arioc: High-concurrency short-read alignment on multiple GPUs

Richard Wilton[1]*, Alexander S. Szalay[1,2]

**1** Department of Physics and Astronomy, Johns Hopkins University, Baltimore, Maryland, United States of America, **2** Department of Computer Science, Johns Hopkins University, Baltimore, Maryland, United States of America

* richard.wilton@jhu.edu

## Abstract

In large DNA sequence repositories, archival data storage is often coupled with computers that provide 40 or more CPU threads and multiple GPU (general-purpose graphics processing unit) devices. This presents an opportunity for DNA sequence alignment software to exploit high-concurrency hardware to generate short-read alignments at high speed. Arioc, a GPU-accelerated short-read aligner, can compute WGS (whole-genome sequencing) alignments ten times faster than comparable CPU-only alignment software. When two or more GPUs are available, Arioc's speed increases proportionately because the software executes concurrently on each available GPU device. We have adapted Arioc to recent multi-GPU hardware architectures that support high-bandwidth peer-to-peer memory accesses among multiple GPUs. By modifying Arioc's implementation to exploit this GPU memory architecture we obtained a further 1.8x-2.9x increase in overall alignment speeds. With this additional acceleration, Arioc computes two million short-read alignments per second in a four-GPU system; it can align the reads from a human WGS sequencer run–over 500 million 150nt paired-end reads–in less than 15 minutes. As WGS data accumulates exponentially and high-concurrency computational resources become widespread, Arioc addresses a growing need for timely computation in the short-read data analysis toolchain.

This is a *PLOS Computational Biology* Software paper.

## Introduction

Short-read DNA sequencing technology has been estimated to generate 35 petabases of DNA sequencer data per year [1], with the amount of new data increasing exponentially [2]. Much of this data resides in commercial and academic repositories that may contain thousands or tens of thousands of DNA sequencer runs. Copying this quantity of data remotely is costly and time-consuming. For this reason, it can be more practical to analyze and reduce the data without transferring it out of the repository where it is archived. As a consequence, high-throughput computational resources are becoming an integral part of the local computing environment in data centers that archive sequencing data.

**Funding:** The author(s) received no specific funding for this work. This work used the Extreme Science and Engineering Discovery Environment (XSEDE resource BRIDGES GPU-AI, allocation CCR190056), which is supported by National Science Foundation grant number ACI-1548562. Specifically, it used the Bridges system, which is supported by NSF award number ACI-1445606, at the Pittsburgh Supercomputing Center (PSC). Neither XSEDE nor PSC had any role in study design, data collection and analysis, decision to publish, or preparation of the manuscript.

**Competing interests:** The authors have declared that no competing interests exist.

A fundamental step in analyzing short-read DNA sequencer data is read alignment, the process of establishing the point of origin of each sequencer read with respect to a reference genome. Read alignment is algorithmically complex and time-consuming. It can represent a bottleneck in the prompt analysis of rapidly accumulating sequencer data.

A fruitful approach to improving read-aligner throughput is to compute alignments using the parallel processing capability of general-purpose graphics processing units, or GPUs. GPU-accelerated read-alignment software such as Arioc [3, 4] and SOAP3-dp [5] can provide order-of-magnitude speed increases compared with CPU-only implementations such as BWA-MEM [6], Bowtie 2 [7], and Bismark [8]. In recent years, the speed of CPU-only implementations has increased with improvements in CPU speed, multithreaded concurrency, and hardware memory management, but GPU-accelerated implementations also run faster on newer GPUs that support a greater number of GPU threads and higher-bandwidth memory-access capabilities (S1 Table).

### Increased speed due to more capable GPU memory architecture

In this regard, Arioc benefits specifically from larger GPU device memory and high-bandwidth peer-to-peer (P2P) memory-access topology among multiple GPUs [9, 10]. This is because Arioc's implementation relies on a set of large in-memory lookup tables (LUTs) to identify candidate locations in the reference genome sequence at which to compute alignments. For example, these LUTs can occupy up to 77GB for the human reference genome.

When GPU memory is insufficient to contain the LUTs, Arioc places them in page-locked system RAM that the GPU must access across the PCIe bus. When GPU memory is large enough to contain these tables and P2P memory interconnect is supported, LUT data accesses execute 10 or more times faster and the overall speed of the software increases accordingly.

### Lookup-table layouts adapted to available GPU features

Arioc can be configured to use one of three memory-layout configurations for its lookup tables (Fig 1) according to the total amount of installed GPU memory and the availability of GPU P2P memory access:

- LUTs in page-locked system RAM, mapped into the GPU address space

- LUTs duplicated in each available GPU

- LUTs partitioned across all available GPUs

A lookup-table layout in page-locked system RAM relies on slower memory accesses through the PCIe bus; a layout where the smaller of the LUTs is copied to each GPU instance uses page-locked system RAM only for the larger LUTs; and a fully-partitioned layout places the LUT data entirely in GPU memory where data-access speed is greatest.

### Software design and implementation

The Arioc aligner is written in C++. The compiled program runs in one computer with one or more Nvidia GPUs and on a minimum of two concurrent CPU threads per GPU. The program uses 38 different CUDA kernels written in C++ (management and prioritization of alignment tasks, alignment computation) and about 150 calls to various CUDA Thrust APIs (sort, set reduction, set difference, string compaction).

The software is implemented as a pipeline through which batches of reads are processed [3]. For each batch of reads, Arioc computes alignments in parallel on each available GPU

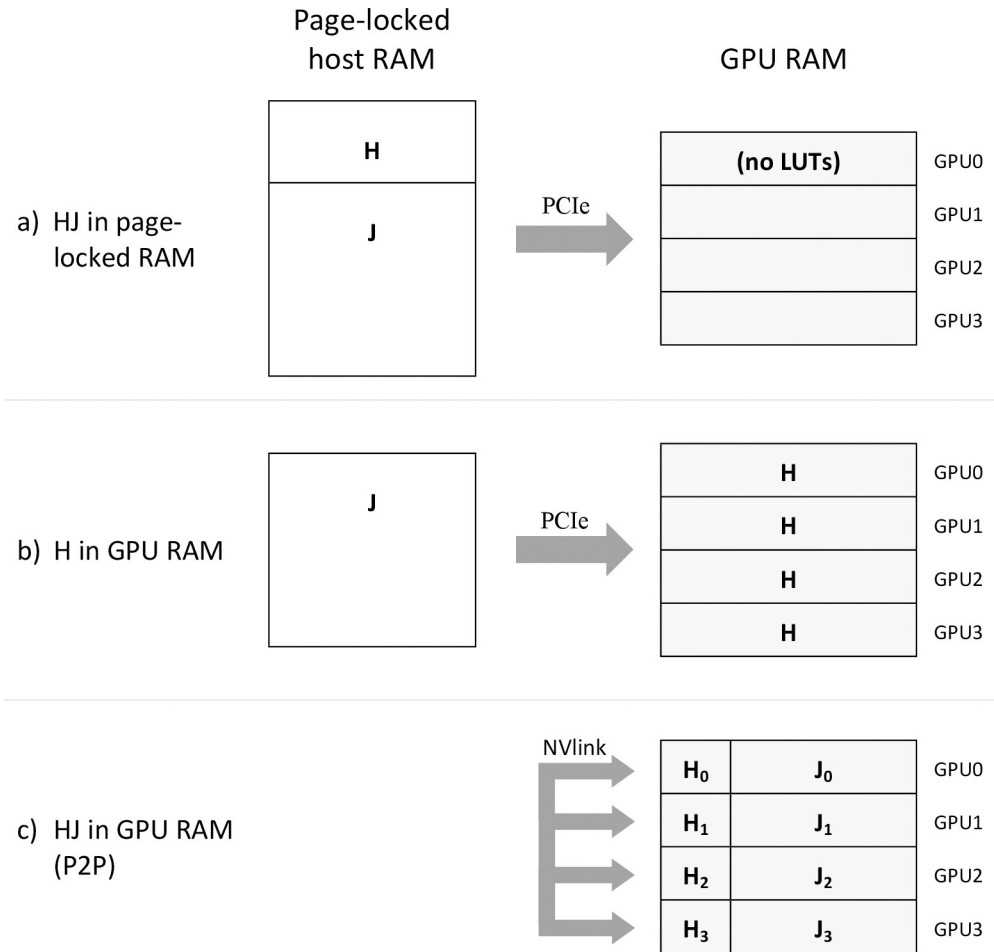

**Fig 1. Lookup table memory layouts supported by Arioc.** The H lookup table is a hash table that contains J-table offsets; the J table contains reference-sequence locations. (a) H and J tables both reside in page-locked system RAM; all memory accesses from the GPU traverse the PCIe bus. (b) A copy of the H table resides in device RAM on each GPU; only J-table data traverses the PCIe bus. (c) The H and J tables are partitioned across device RAM on all available GPUs; GPU peer-to-peer memory accesses use the NVlink interconnect.

device, so the number of reads in a batch is limited by available GPU memory. The software uses concurrently-executing CPU threads for disk I/O and for computing per-read metadata, including alignment scores, mapping quality scores, and methylation context for WGBS reads, in SAM format [11].

The current version of Arioc uses the same basic alignment algorithms as were implemented in earlier versions [3]: a fast spaced-seed nongapped alignment implementation for reads with few differences from the reference sequence, and a slower gapped-alignment implementation for reads with multiple differences from the reference. The accuracy of the current version was validated against SOAP3-dp and Bowtie 2 using simulated human short-read data (S5 Text, S1 Data) at error rates of 0.25% (to approximate typical Illumina sequencer error rates [12]) and 7.0% (to simulate read sequences with multiple differences from the reference genome).

## Results

We assessed Arioc's performance on three different computers provisioned with multiple Nvidia V100 GPUs (Table 1). The computer systems represented three different operating

**Table 1. Computers used for Arioc performance testing.** (See S1 Text for GPU peer-to-peer memory topology).

| computer | CPU threads<br>system RAM | GPUs<br>GPU RAM | GPU interconnect |
|---|---|---|---|
| Dell PowerEdge C4140[a]<br>2 x Intel Xeon Gold 6148 | 40@2.4GHz<br>384GB | 4 x Nvidia V100<br>32GB | NVlink 2.0 (SXM2) |
| AWS p3dn.24xlarge instance[b]<br>2 x Intel Xeon Platinum 8175M | 96@2.5GHz<br>768GB | 8 x Nvidia V100<br>32GB | NVlink 2.0 (SXM2) |
| Nvidia DGX-2[c]<br>2 x Intel Xeon Platinum 8168 | 96@2.7GHz<br>1.5TB | 16 x Nvidia V100<br>32GB | NVlink 3.0 (SXM3)<br>+ NVswitch |

[a]Dell EMC [13].

[b]AWS [14].

[c]PSC [15, 16].

environments: an experimental high-performance computing cluster with exclusive single-user access to a single machine, an academic research cluster with shared access to computing and network resources, and a commercial cloud environment. To avoid contention from other concurrently executing applications for CPU, GPU, or disk I/O resources, we measured performance only when no other user programs were executing, and we used local (as opposed to network) filesystems for output.

Since LUT accesses represent a significant portion of Arioc's execution time, we measured the extent to which the use of GPU P2P interconnect hardware affects Arioc's overall speed. We did this by comparing the speed of each of the three LUT-layout patterns supported by the software. In each computer, we ensured that hardware and CUDA driver software were configured for GPU P2P memory access (S1 Text). On computers whose GPU P2P topology did not include all possible GPU pairs, we ran Arioc only on a subset of GPU devices that supported mutual P2P interconnect.

We used Arioc with human reference genome release 38 [17] to align four public-domain paired-end whole genome sequencer runs (Table 2) and filtered the results to exclude all but proper mappings, that is, alignments that were concordant with criteria for Illumina paired-end mappings (forward-reverse orientation of the mates, inferred fragment length no greater than 500). Each of the four sequencer runs provided at least 40x sequencing coverage, but each differed from the others in at least one of the following characteristics:

- read length: 100nt or 150nt

- sequencing technology: WGS or WGBS

- percentage of reads with proper (concordant) mappings: below 80% or above 90%

**Table 2. Whole genome sequencing (WGS) and whole genome bisulfite sequencing (WGBS) runs used for Arioc performance testing.**

| sample:run | type | pairs<br>(reads) | read length | properly mapped |
|---|---|---|---|---|
| ERP010710:ERR1347703[a] | WGS | 681,380,865 (1,362,761,730) | 2×100nt | 79.31% |
| ERP010710:ERR1419128[a] | WGS | 596,611,242 (1,193,222,484) | 2×100nt | 96.56% |
| SRP117159:SRR6020687[b] | WGBS | 534,647,118 (1,069,294,236) | 2×150nt | 89.95% |
| SRP117159:SRR6020688 | WGS | 419,380,558 (838,761,116) | 2×150nt | 96.52% |

[a]See [18].

[b]See [19].

To determine the speed-versus-sensitivity pattern for the alignment of each sequencer run, we recorded speed (overall throughput) and sensitivity for a variety of settings of *maxJ*, a run-time parameter that specifies the maximum number of candidate reference-sequence locations per seed. When Arioc is configured with higher *maxJ* settings, speed decreases and sensitivity increases (S2 Text).

We carried out speed-versus-sensitivity experiments using all four sets of sequencer reads and with all three LUT layouts in GPU memory. We characterized speed (throughput) as the number of read sequences (paired-end mates) per second for which the software computes alignments. Sensitivity was recorded as the percentage of the pairs for which the aligner reported at least one proper (concordant) mapping.

Overall, aligner speed varied by a factor of 10 across all the short-read data and all the hardware and software configurations we tested, with wide variations attributable to the characteristics of the read-sequence data as well as to the runtime software configuration. With each whole-genome sequencing sample, speed always decreased with increasing sensitivity, with a pronounced drop-off in speed as the aligner approaches maximum sensitivity.

## Computing environments

With a four-GPU computer in a high-performance computing research cluster [13] across all samples and speed-versus-sensitivity settings, the maximum throughput was 2,195,046 reads/second for 100nt paired-end reads and 1,686,477 reads/second for 150nt paired-end reads. On a commercial cloud compute instance [14], speeds were generally about 10% slower. With an Nvidia DGX-2 computer in an academic research cluster [20], speeds were generally 5–10% faster (S1 Data).

## GPU memory

Speed increased in proportion to the amount of LUT data residing in GPU memory as opposed to page-locked system memory (Fig 2). The relative increase in speed varied from about 1.5 to about 2.5. This was most apparent when Arioc was parameterized for maximum speed and lower sensitivity.

## Read-sequence characteristics

In general, throughput for 150nt WGBS reads was about 2/3 of the throughput for 150nt WGS reads (S1 Data). With 150nt WGS reads, throughput was about 3/4 of the throughput for 100nt WGS reads. There was no evident relationship between throughput and the percentage of proper (concordant) mappings.

## Scaling with additional GPUs

Throughput increased when additional GPU devices were available (S1 Data). Scaling was nearly ideal with 8 or fewer GPUs. With 9 or more GPUs, speed continued to increase but the gain in speed was proportionally smaller with each additional GPU.

## Disk write bandwidth

With multiple GPUs, Arioc runs up to 10% faster when writing output to a filesystem on a dedicated high-bandwidth disk device than when using a network filesystem (S1 Data). The speed increase attributable to higher-performance disk storage was greater when Arioc was configured for higher speed and lower sensitivity.

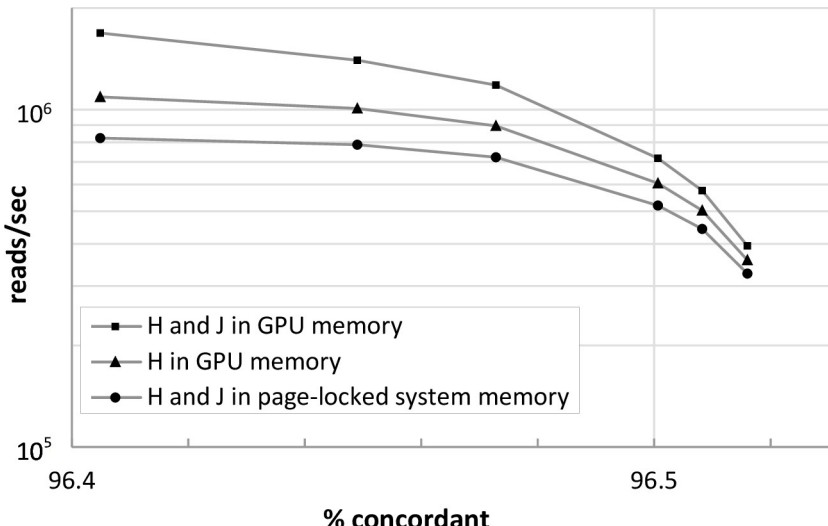

**Fig 2. Speed versus sensitivity for three GPU memory-layout techniques.** Speed (reads/second) is greater when GPU peer-to-peer memory interconnect is used, for a range of sensitivity (% concordant) settings. Speeds are highest when the H and J tables are partitioned across device RAM on all available GPUs and GPU peer-to-peer memory accesses use the direct P2P memory interconnect. Speeds are lower with H in device RAM on each GPU and J in page-locked system memory. Speeds are lowest when H and J both reside in page-locked system RAM. H table: 25GB; J table: 52GB. Data from SRR6020688 (S1 Data).

## Comparison with CPU-only software

We performed WGS and WGBS speed-versus-sensitivity comparisons with a widely-used WGS aligner, Bowtie 2 [7], and with a well-known WGBS aligner, Bismark [8] (S1 Data). These observations confirmed the order-of-magnitude difference in speed that we had observed in previous experiments [3] [4]. We also estimated system memory utilization with each aligner (S6 text). Arioc uses more system memory to contain reference-genome index structures than do CPU-only implementations; its maximal memory utilization was about 1/3 of available system memory in the most conservatively provisioned computers we used.

## Advantages and limitations

The most important element in the design of the Arioc software is that it uses algorithms and implementation techniques that are amenable to GPU acceleration. This design approach is validated by the data throughputs we have observed on computers provisioned with four or more GPU devices. With Arioc, using GPU P2P memory interconnect leads to higher speeds throughout the usable range of speed-versus-sensitivity configurations. The speed increase is greater with commonly used runtime parameterizations that favor higher speed over sensitivity.

With a four-GPU system, Arioc can align the reads from a human whole-genome sequencing run with 40x coverage in less than 15 minutes. However, although Arioc supports reference genomes up to $2^{34}$ base pairs (17G base pairs) in size, the lookup tables for such genomes are proportionately larger. The use of a partitioned LUT memory configuration to obtain maximum speed for a given reference genome thus depends on the size of the LUTs for that genome as well as the amount of interconnected GPU memory.

High-concurrency GPU interconnect architectures are increasingly available not only in dedicated high-performance computing environments but in also academic or commercial computing clusters where they are typically used as shared resources. In the latter case,

however, maximum throughput is obtainable only when P2P topology mutually interconnects all the GPUs on which Arioc executes and when no other concurrently executing programs can contend for GPU memory-access bandwidth.

We observed, in both the WGS and the WGBS samples, the same pattern of decreasing speed with increasing sensitivity. With maximum sensitivity, all short-read aligners display a sharp fall-off in speed due to the amount of work performed in searching for mappings for "hard to map" read sequences that contain multiple differences from the reference genome, but the speed-versus-sensitivity pattern provides insight into the behavior of each of the aligners we evaluated and suggests an optimal choice of speed versus sensitivity for a given data sample. For Arioc in particular, the range of possible speeds depended on how many GPUs were used (that is, the maximum number of concurrently executing GPU threads), the layout of lookup tables in GPU memory, the sequencing technology (WGS or WGBS), and read length.

Although alignment speeds improve with additional GPUs, there may be limited practical value in using more than four GPUs concurrently. When alignment throughput exceeds one million reads per second, factors other than computation speed become increasingly important. These may include the time required to initialize GPU lookup tables, data-transfer speeds between CPU and GPU memory, available GPU interconnect bandwidth, and available disk I/O bandwidth.

A common approach to aligning multiple WGS samples is to use a CPU-only aligner such as BWA-MEM or Bowtie 2 on dozens or hundreds of CPU cores, either by splitting the work across multiple computers or by using systems that support a large number of CPU threads [21]. Our results suggest that using Arioc in a single multi-GPU computer is more effective, in terms of both sensitivity and of throughput, than either CPU-only strategy.

Any short-read analysis protocol could benefit from this kind of read-aligner performance, but the potential time and cost savings would be substantial in a large-scale repository of DNA sequencing data. In recent years an increasing number of large WGS datasets have become available in shared-access data centers and in the commercial cloud [22, 23, 24, 25, 26]. In these computing environments, high-throughput short-read alignment is likely to be carried out on every WGS sample, either as an initial step in a short-read alignment analysis pipeline or as preparation for data archival.

When we compared Arioc performance with CPU-only short-read aligners where a monetary price is associated with computational resource utilization, Arioc costs half as much and executes ten or more times faster (Fig 3). Although hardware-resource sharing in a cloud environment degrades software performance [27], and pricing for cloud-based computing services, data storage [28], and data transfer [29] is complex and may fluctuate, our results imply that Arioc has significant potential for rapidly and economically aligning large aggregates of short-read sequencer data in a commercial cloud setting.

## Future directions and availability

Improving the performance of one component in a pipelined application inevitably accentuates potential bottlenecks in other components of the pipeline. For example, alleviating the bottleneck related to GPU memory management in Arioc emphasized the inverse relationship of read length and throughput. Arioc's seeding strategy is a simple divide-and-conquer procedure that starts with examining the set of contiguous seeds that span the read sequence. With longer read sequences, a strategy of initially examining fewer seeds might increase throughput without sacrificing sensitivity.

Ideally, Arioc's throughput would increase in direct proportion to the number of available GPU devices. That it does not is most likely related to a limitation in memory bandwidth,

**WGS**

| | % concordant | elapsed time | cost |
|---|---|---|---|
| **Arioc** | 96.40 | 0:13:03 | $6.79 |
| | 96.45 | 0:14:33 | 7.57 |
| | 96.47 | 0:16:14 | 8.45 |
| | 96.50 | 0:23:45 | 12.36 |
| | 96.51 | 0:28:33 | 14.85 |
| | 96.52 | 0:39:04 | 20.33 |
| **Bowtie 2** | 94.61 | 5:29:31 | 12.65 |
| | 94.98 | 6:51:14 | 15.79 |
| | 95.28 | 10:21:38 | 23.87 |
| | 95.30 | 18:22:33 | 42.34 |

**WGBS**

| | % concordant | elapsed time | cost |
|---|---|---|---|
| **Arioc** | 88.58 | 0:22:10 | $11.53 |
| | 89.07 | 0:30:35 | 15.91 |
| | 89.42 | 0:31:40 | 16.47 |
| | 89.69 | 0:40:53 | 21.27 |
| | 89.82 | 0:49:19 | 25.65 |
| | 89.95 | 1:07:09 | 34.93 |
| **Bismark** | 85.54 | 25:01:05 | 57.64 |

**Fig 3. Sensitivity (as overall percentage of concordantly mapped pairs), overall elapsed time, and dollar cost for WGS and WGBS alignment on Amazon Web Services virtual machine instances.** WGS results for SRR6020688 (human, 419,380,558 150nt pairs); WGBS results for SRR6020687 (human, 534,647,118 150nt pairs). Arioc: EC2 p3dn.24xlarge instance ($31.212/hour). Bowtie 2, Bismark: EC2 m5.12xlarge instance ($2.304/hour).

either in the transfer of data between CPU-addressable memory and on-device GPU memory or in randomly accessing large amounts of data in GPU memory. In either case, achieving near-ideal scaling beyond 8 GPUs would require nontrivial software re-engineering to further decrease the amount of data transferred between CPU and GPU and to optimize the utilization of GPU memory bandwidth [9].

Short-read sequencing technology remains an essential step in DNA sequence analysis on the petabyte and exabyte scale. In an analysis workflow that includes software such as samtools [30], Picard tools [31], or Bismark tools [8], short-read alignment may represent a comparatively small proportion of the time required to complete the workflow. For this reason, a number of software tools are being developed [32, 33] that use high-concurrency computing hardware to process large quantities of WGS and WGBS data. The full potential for rapid analysis of WGS samples with Arioc may eventually be realized only when the entire software toolchain utilizes concurrent CPU and GPU resources.

Arioc's current implementation is fast, but advances in hardware and in software technology will inevitably lead to even faster implementations in the future. Nevertheless, given the trend toward associating high-concurrency CPU and GPU hardware with large repositories of whole genome sequencing data, Arioc can fill the need for a high-throughput general-purpose short-read alignment tool that exploits the capabilities of the hardware.

The Arioc software is available at https://github.com/rwilton/arioc. It is released under a BSD open-source license.

## Supporting information

**S1 Text. GPU peer-to-peer memory interconnect topology.**
(DOCX)

**S2 Text. Arioc configuration parameters for WGS and WGBS alignments.**
(DOCX)

**S3 Text. Bowtie 2 configuration parameters for WGS alignments.**
(DOCX)

**S4 Text. Bismark configuration parameters for WGBS alignments.**
(DOCX)

**S5 Text. Correct vs. incorrect mappings, classified by MAPQ.**
(DOCX)

**S6 Text. Arioc hardware resource utilization.**
(DOCX)

**S1 Table. Representative Nvidia GPU devices since 2012.**
(DOCX)

**S2 Table. Short-read aligner versions and distribution websites.**
(DOCX)

**S1 Data • Aggregated data for Arioc and Bowtie 2 (Dell EMC HPC and Innovation Lab) • Aggregated data for Arioc, Bowtie 2, and Bismark (Amazon Web Services EC2) • Aggregated data for Arioc (Pittsburgh Supercomputing Center) • Throughput with multiple GPUs • Effect of disk filesystem architecture on Arioc throughput • Correct vs incorrect mappings, classified by MAPQ.**
(XLSX)

## Acknowledgments

We are grateful to Steven Salzberg and Ben Langmead for suggesting that high-speed short-read alignment could contribute to the cost-effectiveness of managing sequencer read data in a cloud environment; to Joseph Stanfield and John Haag at the Dell EMC HPC and AI Innovation Lab for access to the Rattler test/development cluster; to Sanjay Padhi at Amazon Web Services for facilitating access to the AWS Cloud Credits for Research program; to Jaime Combariza, Ryan Bradley, and Matthew Wang for help and support at the Maryland Advanced Research Computing Center (MARCC); to Anthony Kolasny for helping to arrange access to XSEDE resources; to Roberto Gomez and TJ Olesky for help and support at the Pittsburgh Supercomputing Center (PSC).

## Author Contributions

**Conceptualization:** Richard Wilton.

**Data curation:** Richard Wilton.

**Formal analysis:** Richard Wilton.

**Funding acquisition:** Alexander S. Szalay.

**Investigation:** Richard Wilton, Alexander S. Szalay.

**Methodology:** Richard Wilton.

**Project administration:** Richard Wilton, Alexander S. Szalay.

**Resources:** Richard Wilton, Alexander S. Szalay.

**Software:** Richard Wilton.

**Supervision:** Alexander S. Szalay.

**Validation:** Richard Wilton.

**Visualization:** Richard Wilton, Alexander S. Szalay.

**Writing – original draft:** Richard Wilton.

**Writing – review & editing:** Richard Wilton, Alexander S. Szalay.

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
