## [Decision Letter · Decision Letter 0]

7 Apr 2020

Dear Dr Wilton,

Thank you very much for submitting your manuscript "Arioc:  high-concurrency short-read alignment on multiple GPUs" for consideration at PLOS Computational Biology.

As with all papers reviewed by the journal, your manuscript was reviewed by members of the editorial board and by several independent reviewers. In light of the reviews (below this email), we would like to invite the resubmission of a significantly-revised version that takes into account the reviewers' comments.

We cannot make any decision about publication until we have seen the revised manuscript and your response to the reviewers' comments. Your revised manuscript is also likely to be sent to reviewers for further evaluation.

Sincerely,

Manja Marz

Software Editor

PLOS Computational Biology

Manja Marz

Software Editor

PLOS Computational Biology

Reviewer's Responses to Questions

**Comments to the Authors:**

Reviewer #1: Authors present a new implementation of their short-read aligner Arioc.

The proposed implementation takes advantage of the most recent GPU hw architecture supporting high-bandwidth P2P memory access among multiple GPUs to increase its performance in terms of computing time.

Although the work is interesting there are several points that need to be clarified.

Major comments

1. My main concern is related to the results.

1.a - Authors did not perform experiments on simulated libraries with the aim to assess the reliability of the new version of ARIOC. These analyses were carried out with previous versions of the tool. However, as this is a new implementation it must be validated. Authors should perform these experiments and compare the results with those of the other tools. Simulated libraries should also be permanently accessible through a DOI or according to the journal policies.

1.b - Experiments on real data are not complete. For instance, for Bowtie2 results were only reported for the library SRR6020688. Experiments must be performed for all libraries.

1.c - Unbalanced hardware configurations were used to analyze the performance ARIOC and the other tools. For instance using AWS (Suppl Data D2) ARIOC was run using 4 NVIDIA V100 and 96 threads whereas Bowtie 2 was run using 40 threads. Experiments should be performed using an identical hardware configuration and using modern processors supporting hundreds of cores. A comparison between Bowtie2/Bismark and ARIOC with a single GPU and identical modern processor should also be performed.

1.d - As done for Bowtie2 and Bismark comparative experiments should also be performed for SOAP3-dp with the aim to confirm the best performances of ARIOC. As a reader, I would be curios to read about the behaviour of SOAP3-dp with the new V100 equipped with 32GB of memory. Also in this case experiments should be performed using an identical hardware configuration.

1.e - Results in supplementary data (D1 - D5) should be report the same (common) information for all tools. For instance, the table reporting results for Bowtie2 (Suppl Data D2) reports a column labeled “overall mapped %”, but the same column is not present in tables for Arioc. Moreover, for an easy reading of the results, each column of the tables should be described.

1.f - In Supplementary Data D1 are reported performance obtained for an unpublished library “LIBD1373”. I don't think the library is mentioned in the article nor is it downloadable from the ARIOC repository at https://github.com/rwilton/arioc.

1.g - Figure 4 show speed vs sensitivity for the LUT layouts. I observe that increasing the sensitivity (par. maxJ) the speed tends to converge for all LUT layouts. It would seem that by increasing the sensitivity there are not more advantages related to the device memory and NVLINK.

1.h - Can you also comment about the overall host (and device) memory consumption? Is the host (device) memory consumption comparable with Bowtie2/Bismark/SOAP3-dp?

2. All experiments were carried out with the human genome. Today the scientific community is also heavily involved in the study of more complex genomes. My question is whether ARIOC and its LUT are suitable for very big and highly repetitive genomes.

3. Line 41: “The first step in analyzing short-read DNA sequencer data is read alignment …” It would be more correct to write that alignment is at the base of the NGS analyses. Quality control and filtering are mandatory steps before alignment.

4. Finally, I suggest (it is not mandatory) to dockerize the tool installation and deposit it on DockerHub and/or BioContainers. Containers allow a fast deployment on local clusters as well as on the cloud.

Minor comments

Labels “Figure 2” and “Figure 3” refer to tables.

Line 58: (For example, these LUTs can occupy up to …): remove the brackets

Reviewer #2: This manuscript addresses an important area of short-read alignment. A new version of Arioc has been developed by utilising the multiple GPUs on the same machine to achieve a substantial speed up in short-read alignment. This new version of Arioc can benefit from larger GPU memories and high-bandwidth peer-to-peer memory access between GPUs. The manuscript is well written. I have a few comments:

The authors should explicitly mention that this is a newer version of Arioc.

According to the results, Arioc has a better performance compared with the other aligners like Bowtie2 (for WGS) and Bismark (for WGBS) in terms to accuracy and running time. It is understandable that Arior performs faster due to the utilisation of multiple GPU technology. However, why does Arior have a better accuracy? Is it due to the algorithm used in Arioc? The authors should briefly explain this in the manuscript.

The authors need to explain what "concordant" alignment means. This is important in order to understand how the accuracies of different aligners were evaluated.

Figure 5 mentions "concordantly-mapped pairs". Does this figure only consider the aligned pairs? How about the case that one of reads in the pair is properly aligned but another cannot? If Figure 5 only considers the aligned pairs, it is necessary to have another figure showing, for different aligners, the % of reads concordantly aligned, including those either with only one read or both reads in the pairs being aligned.

This manuscript does not mention about the alignment algorithm used in Arioc. The authors should briefly mention this and refer to their previous papers for details.

**Have all data underlying the figures and results presented in the manuscript been provided?**

Reviewer #1: None

Reviewer #2: Yes

PLOS authors have the option to publish the peer review history of their article (what does this mean?). If published, this will include your full peer review and any attached files.

Reviewer #1: No

Reviewer #2: No
---

## [Decision Letter · Decision Letter 1]

30 Jul 2020

Dear Dr Wilton,

Thank you very much for submitting your manuscript "Arioc:  high-concurrency short-read alignment on multiple GPUs" for consideration at PLOS Computational Biology.

As with all papers reviewed by the journal, your manuscript was reviewed by members of the editorial board and by several independent reviewers. In light of the reviews (below this email), we would like to invite the resubmission of a significantly-revised version that takes into account the reviewers' comments.

We cannot make any decision about publication until we have seen the revised manuscript and your response to the reviewers' comments. Your revised manuscript is also likely to be sent to reviewers for further evaluation.

Sincerely,

Manja Marz

Software Editor

PLOS Computational Biology

Manja Marz

Software Editor

PLOS Computational Biology

Reviewer's Responses to Questions

**Comments to the Authors:**

Reviewer #1: Although the authors have worked to improve the manuscript, I must point out that some of my indications have not been fully implemented.

In the following I report my comments to the authors' responses.

1.a The authors performed tests on synthetic data as required. However, they did not provide a link to download the simulated libraries as required. Readers must be able to reproduce the experiments. Please, provide a DOI to share the libraries with the readers.

1.c Considerations made by the authors are interesting and should be reported in the manuscript. However a comparison of the performance in terms of computing time must be based on an identical hw configuration. I have no doubt that ARIOC outperforms the other tool, but what is the information given by performances obtained with unbalanced hw configuration for different tools?

1.g Please report in the manuscript the considerations given in your reply to the author.

1.h As requested, the authors report a comparison in terms of memory consumption between ARIOC and the other tools. The results of the comparison are reported in Appendix A6. However, no discussion of memory consumption is provided in the manuscript as well as no reference is made to Appendix A6. Please, discuss in the manuscript about the performances in terms of memory consumption. What considerations about the high RAM required by ARIOC with respect to the other tools?

2. Please report in the manuscript the considerations given in your reply to the author. Constraints as the maximum genome size as well as other limitations to use ARIOC must be reported and discussed in the manuscript. This information is very important for researchers that plan to use a tool.

4. As wrote in my previous comment, I only suggested to dockerize ARIOC. I can understand the difficulties of building an optimized container. In any case, using your Ferrari/Volkswagen comparison, observe that only few people drive a Ferrari whereas a lot drive a Volkswagen. Since the installation and configuration take hours, the risk is that an unskilled user will abandon the tool in favour of others that are easy to install. For the future, I suggest to take into account the possibility to build a container for ARIOC.

**Have all data underlying the figures and results presented in the manuscript been provided?**

Reviewer #1: None

PLOS authors have the option to publish the peer review history of their article (what does this mean?). If published, this will include your full peer review and any attached files.

Reviewer #1: No
---

## [Decision Letter · Decision Letter 2]

10 Sep 2020

Dear Dr Wilton,

We are pleased to inform you that your manuscript 'Arioc:  high-concurrency short-read alignment on multiple GPUs' has been provisionally accepted for publication in PLOS Computational Biology.

Best regards,

Manja Marz

Software Editor

PLOS Computational Biology

Reviewer's Responses to Questions

**Comments to the Authors:**

Reviewer #1: The authors properly addressed all comments

**Have all data underlying the figures and results presented in the manuscript been provided?**

Reviewer #1: None

PLOS authors have the option to publish the peer review history of their article (what does this mean?). If published, this will include your full peer review and any attached files.

Reviewer #1: No

---

## [Editor Report · Acceptance letter]

26 Oct 2020

PCOMPBIOL-D-20-00150R2 

Arioc: High-concurrency short-read alignment on multiple GPUs

Dear Dr Wilton,

I am pleased to inform you that your manuscript has been formally accepted for publication in PLOS Computational Biology. Your manuscript is now with our production department and you will be notified of the publication date in due course.

With kind regards,

Laura Mallard
